# Cortical magnification in human visual cortex parallels task performance around the visual field

**Noah C Benson[1,2‡*], Eline R Kupers[1,2§], Antoine Barbot[1,2#], Marisa Carrasco[1,2†], Jonathan Winawer[1,2†]**

[1]Department of Psychology, New York University, New York, United States; [2]Center for Neural Sciences, New York University, New York, United States

**Abstract** Human vision has striking radial asymmetries, with performance on many tasks varying sharply with stimulus polar angle. Performance is generally better on the horizontal than vertical meridian, and on the lower than upper vertical meridian, and these asymmetries decrease gradually with deviation from the vertical meridian. Here, we report cortical magnification at a fine angular resolution around the visual field. This precision enables comparisons between cortical magnification and behavior, between cortical magnification and retinal cell densities, and between cortical magnification in twin pairs. We show that cortical magnification in the human primary visual cortex, measured in 163 subjects, varies substantially around the visual field, with a pattern similar to behavior. These radial asymmetries in the cortex are larger than those found in the retina, and they are correlated between monozygotic twin pairs. These findings indicate a tight link between cortical topography and behavior, and suggest that visual field asymmetries are partly heritable.

**\*For correspondence:**
nben@uw.edu

[†]These authors contributed equally to this work

**Present address:** [‡]eScience Institute, University of Washington, Seattle, United States; [§]Department of Psychology, Stanford University, Stanford, United States; [#]Spinoza Centre for Neuroimaging, Amsterdam, Netherlands

## Introduction

The human visual system processes the visual world with an extraordinary degree of spatial non-uniformity. At the center of gaze, we see fine details whereas in the periphery acuity is about 50 times lower (*Frisén and Glansholm, 1975*). This eccentricity effect is mediated by a far greater density of cone photoreceptors and retinal ganglion cells (RGCs) for the fovea than periphery, as well as by a correspondingly greater amount of cortical surface area per degree of the visual field (*Rodieck, 1998*).

Importantly, visual performance varies not only across eccentricity, but also as a function of polar angle (*Figure 1*); at matched eccentricity, performance is better along the horizontal than the vertical meridian (horizontal-vertical asymmetry, or 'anisotropy'; HVA) and better along the lower than the upper vertical meridian (vertical meridian asymmetry; VMA) (*Carrasco et al., 2001*; *Cameron et al., 2002*; *Montaser-Kouhsari and Carrasco, 2009*; *Corbett and Carrasco, 2011*; *Abrams et al., 2012*; *Baldwin et al., 2012*; *Greenwood et al., 2017*; *Himmelberg et al., 2020*; *Barbot et al., 2021*). These asymmetries are present in several basic perceptual dimensions, including contrast sensitivity (*Rovamo and Virsu, 1979*; *Kroon and van der Wildt, 1980*; *Robson and Graham, 1981*; *Regan and Beverley, 1983*; *Pointer and Hess, 1989*; *Carrasco et al., 2001*; *Cameron et al., 2002*; *Fuller et al., 2008*; *Corbett and Carrasco, 2011*; *Abrams et al., 2012*; *Baldwin et al., 2012*; *Rosén et al., 2014*; *Himmelberg et al., 2020*; *Purokayastha et al., 2021*) and spatial resolution (*Anderson et al., 1992*; *Mackeben, 1999*; *Altpeter et al., 2000*; *Carrasco et al., 2002*; *Talgar and Carrasco, 2002*; *Montaser-Kouhsari and Carrasco, 2009*; *Wilkinson et al., 2016*; *Greenwood et al., 2017*; *Barbot et al., 2021*).

Variations in performance around the visual field are substantial: they can be as pronounced as that of doubling the stimulus eccentricity (*Carrasco et al., 2001*; *Barbot et al., 2019*;

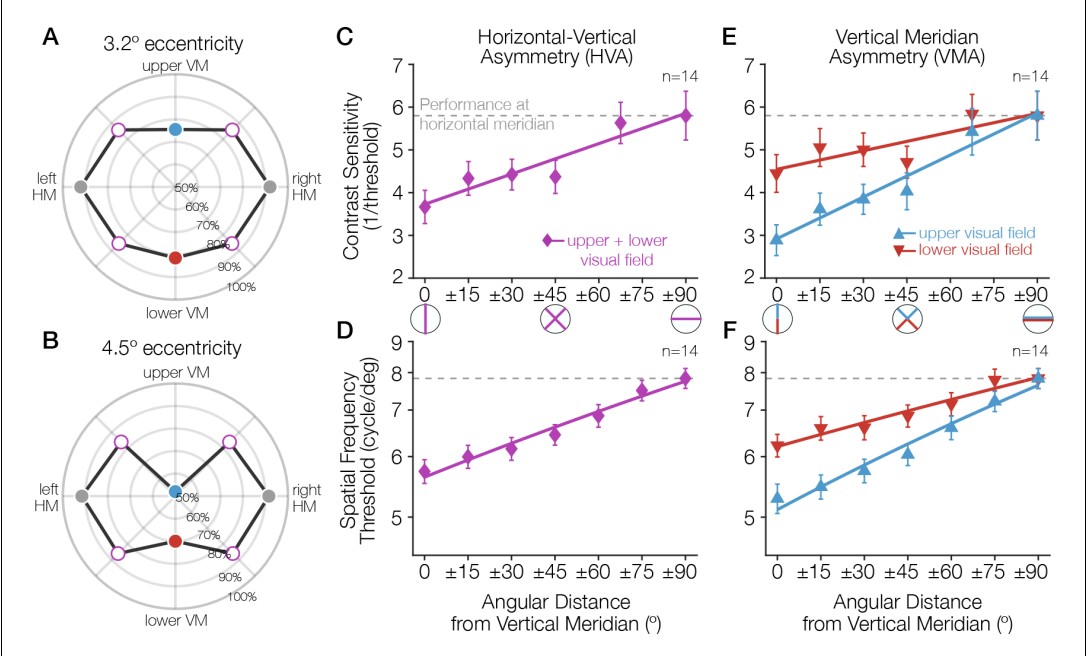

**Figure 1.** Task performance around the visual field. (A, B) Performance for a Gabor-orientation discrimination task is shown in polar coordinates. The plotted angle indicates the polar angle of the tested stimulus location, and the distance from the origin indicates the performance (percent correct) at that polar angle. The origin indicates chance performance. Gabor patches were presented at either (A) 3.2° or (B) 4.5° of eccentricity, with a spatial frequency of 10 and 12 cycles/deg, respectively. The HVA and VMA are evident in both plots and the asymmetries are more pronounced at the farther eccentricity (*Carrasco et al., 2001*). (C–E) As the angular distance from the vertical meridian increases, averaged across upper and lower, performance gradually improves, approaching the performance level at the horizontal (C, D), and the difference in performance between the upper and lower hemifields reduces (E, F). These gradual changes around the visual field have been observed for contrast sensitivity (*Abrams et al., 2012*) (C, E; at 6° eccentricity) and acuity limit (*Barbot et al., 2021*) (D, F; at 10° eccentricity). Each data point corresponds to the average contrast sensitivity (or spatial frequency threshold) at a given angular location, with ±1 SEM across observers. Spatial frequency was measured as the point at which orientation discrimination performance dropped to halfway between ceiling and chance performance. Redrawn using data from *Carrasco et al., 2001*; *Abrams et al., 2012* and *Barbot et al., 2021*. HVA, horizontal-vertical asymmetry; VMA, vertical meridian asymmetry.

*Himmelberg et al., 2020*). Moreover, these asymmetries are in retinal coordinates: the shape of the performance fields are constant in head-centered (but not screen-centered) coordinates whether subjects are tested with their heads tilted or straight (*Corbett and Carrasco, 2011*). This result suggests that the effects likely arise relatively early in the visual processing stream. Consistent with this interpretation, radial asymmetries vary with several low-level stimulus properties: in general, both the HVA and the VMA become more pronounced as target eccentricity, target spatial frequency, or number of distractors increase (*Carrasco et al., 2001*; *Cameron et al., 2002*; *Baldwin et al., 2012*). These radial asymmetries are *not* general hemifield or quarterfield effects; they are largest at the cardinal meridians and decrease gradually as the angular distance from the vertical meridian increases (*Abrams et al., 2012*; *Baldwin et al., 2012*; *Barbot et al., 2021*). Indeed, performance in many tasks is similar across all visual quadrants at 45° intercardinal locations (*Carrasco et al., 2001*; *Cameron et al., 2002*; *Corbett and Carrasco, 2011*; *Abrams et al., 2012*).

The causes of radial asymmetries in performance are largely unknown. In recent work examining this question (*Kupers et al., 2019*; *Kupers et al., 2020*), we simulated a two-alternative forced choice (2-AFC) orientation discrimination task using a computational observer model, implemented in ISETBIO (*Cottaris et al., 2019*), including optical defocus, fixational eye-movements, photoreceptor sampling, and downstream retinal processing with the ISETBIO software platform (*Cottaris et al., 2019*). A linear classifier was trained to predict the 2-AFC response based on the outputs of either the cones or the RGCs from retinal patches varying in cone and RGC density. Variation in the accuracy of the classifier in terms of cell density was small compared to the measured performance differences with polar angle in human observers. Substantial additional radial asymmetries must therefore arise in visual processing efferent from the retina—*i.e.*, the LGN or cortex. Here, we

investigate whether (1) retinotopic maps in human V1 show stark radial asymmetries in cortical surface area; (2) these asymmetries are larger than those found in the retina; (3) these asymmetries fall off gradually with distance from the cardinal meridians, similar to human behavior; and (4) these asymmetries are correlated between monozygotic twins, less correlated between dizygotic twins, and uncorrelated between unrelated (age- and gender-matched) pairs of subjects.

## Results

### Polar asymmetries in cortical surface area

To examine polar angle asymmetries in the cortical representation, we used a dataset of 163 subjects for whom retinotopic maps as well as manually labeled iso-polar angle and iso-eccentricity contours are publicly available. These labeled data (*Benson et al., 2021b*) were produced by annotating the largest publicly available human retinotopy dataset: the Human Connectome Project (HCP) 7 Tesla Retinotopy Dataset (*Benson et al., 2018*). We calculated the surface areas for a series of regions of interest (ROIs) in each subject defined by their visual field preference. Each series of ROIs was centered on one of the four cardinal meridians. For each meridian, we defined the ROI series to cover a gradually increasing span of polar angles from ±10° to ±50° (*Figure 2A*).

Analysis of these ROIs indicates that the cortical surface area in V1 is distributed asymmetrically, with remarkably large differences between the horizontal and vertical meridians. For example, within ±10° of the meridians, the surface area is ~80% greater for the horizontal than the vertical representation (*Figure 2B and D*) and for the lower than the upper meridian (*Figure 2C and E*). Within these ±10° ROIs, 94% of the 163 subjects have positive HVAs, and 93% have positive VMAs. The asymmetries are largest for narrow wedge ROIs. They decrease gradually with angular distance from the vertical meridian, with the VMA decreasing more sharply with distance than the HVA. Previous estimates of V1 cortical magnification by *Horton and Hoyt, 1991* fall about halfway between our measures for the vertical and horizontal meridians as they did not account for angular variation (see Materials and methods). Because the cortical surface area of gyri tends to be large on the pial surface and small on the white-matter surface, whereas the surface area of sulci tends to reverse this trend, we calculated all surface areas on the mid-gray surface. Note, however, that the trends reported above hold for the pial and the white-matter surfaces as well (*Figure 2—figure supplement 1*).

For the VMA estimates, our measures of surface area straddled the V1/V2 boundaries symmetrically, as depicted in *Figure 2A* (right image, red and blue ROIs). We defined the ROIs this way to avoid potential artifacts that can arise from measuring receptive field centers near the vertical meridian, as discussed in prior reports (*Larsson and Heeger, 2006*; *Winawer et al., 2010*). The advantage of this method is that even if the population receptive field (pRF) centers miss the vertical meridian by a few degrees, the boundaries are well defined on both sides of the ROIs (e.g., 10° into V1 and 10° into V2). If we use the V1 portion only, the general pattern of results holds.

### Match between cortical and behavioral asymmetries in distinct subject groups

The cortical patterns we observe match behavioral asymmetries (*Figure 1*), which also decrease gradually away from the vertical meridian, indicating a close correspondence between psychophysical performance and cortical representations in early visual retinotopic maps. Overall, both behavior and cortical surface area fall off gradually and are highly correlated (*Figure 3*; Pearson's $r = 0.99$). This correlation holds for surface areas calculated on the white-matter surface ($r = 0.99$), the mid-gray surface (*Figure 3*; $r = 0.99$), and the pial surface ($r = 0.98$).

Our finding that V1 surface area and acuity threshold covary with polar angle (*Figure 3*) parallels findings that they covary with eccentricity (*Duncan and Boynton, 2003*). Other tasks such as motion sensitivity (*McKee and Nakayama, 1984*), critical flicker frequency (*Hartmann et al., 1979*), texture segmentation (*Gurnsey et al., 1996*; *Barbot and Carrasco, 2017*), and perceptual grouping (*Tannazzo et al., 2014*) do not show the same systematic decline with eccentricity as acuity. For these tasks, performance might be less tightly linked to V1 surface area.

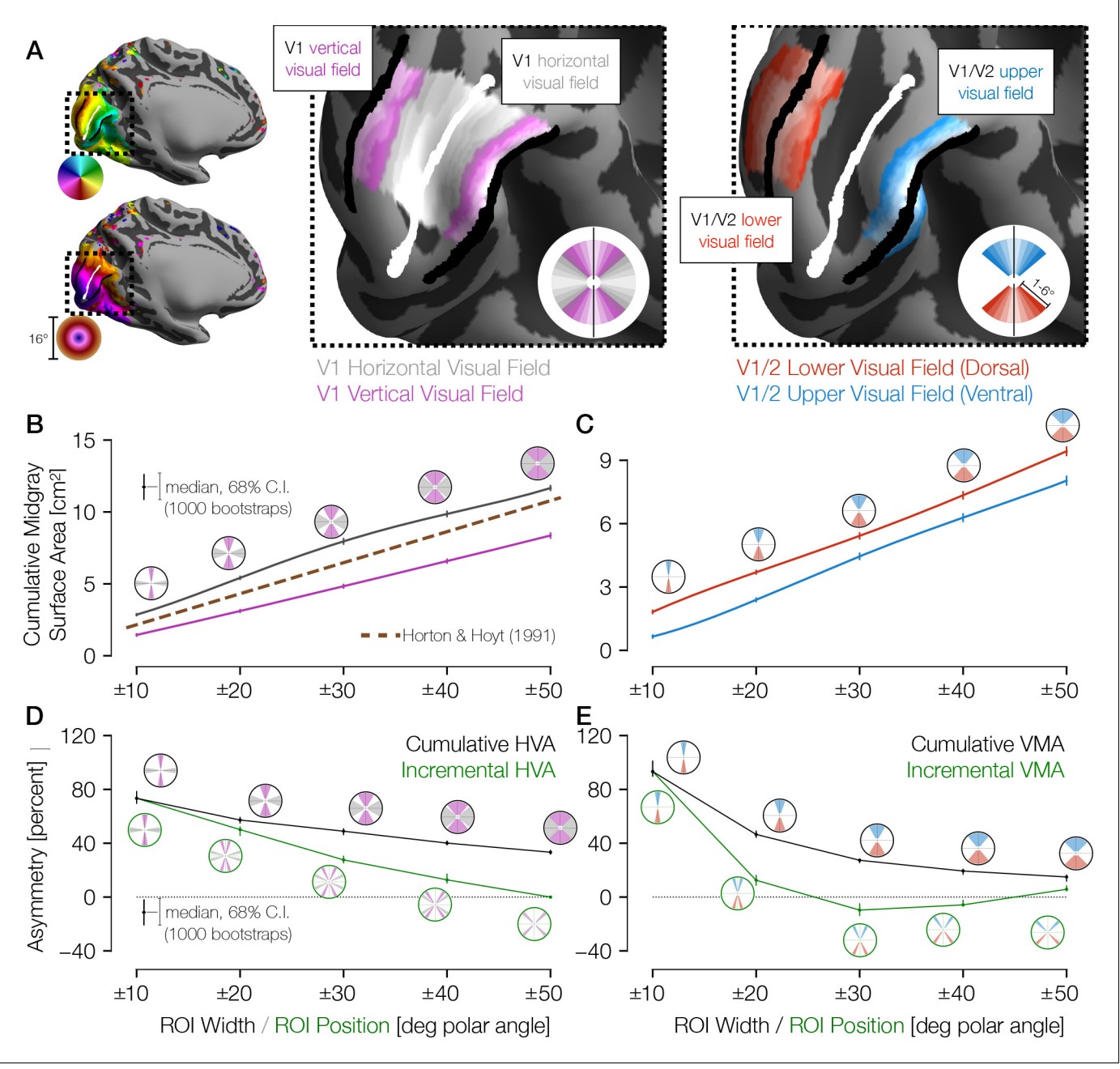

**Figure 2.** Cortical surface areas around the visual field. (A) Polar angle (top left) and eccentricity (bottom left) maps for example HCP subject 177746. The V1/V2 boundaries and the V1 horizontal meridian obtained via Bayesian inference are shown as black and white lines, respectively. The middle panel of **A** shows ROIs drawn with gradually increasing polar angle widths centered around the horizontal meridian (gray) and the vertical meridian (V1/V2 boundary; magenta), limited to 1–6° of eccentricity in V1 only. The right panel shows ROIs around the upper vertical meridian (ventral V1/V2 boundary; blue) and the lower vertical meridian (dorsal V1/V2 boundary; red) that include both V1 and V2. In this hemisphere, the cortical surface area is greater near the horizontal than the vertical meridian and near the lower than the upper vertical meridian. (B) Mid-gray surface area for increasingly large ROIs centered on the vertical (magenta) or horizontal (gray) meridian. The error bars indicate the 68% confidence intervals (CIs) from bootstrapping. The circular icons show the visual field representations of the ROIs for the nearby data points. The x-values of data points were slightly offset in order to facilitate visualization. The brown dotted line shows the equivalent V1 ROI surface area as predicted by *Horton and Hoyt, 1991*. (C) Same as B, but for upper (blue) and lower (red) vertical meridians. (D, E) The surface areas are transformed to percent asymmetry, both for the cumulative ROIs (black) and incremental ROIs (green). Whereas cumulative ROIs represent a wedge of the visual field within a certain polar angle distance $\pm\theta$ of a cardinal axis, incremental ROIs represent dual 10°-wide wedges in the visual field a certain polar angle distance $\pm\theta$ from a cardinal axis. The percent asymmetry of x to y is defined as $100\times(x-y)/\text{mean}(x, y)$. Positive values indicate greater area for horizontal than vertical (D) or for lower than upper regions (E). ROI, region of interest.

*Figure 2 continued on next page*

*Figure 2 continued*

The online version of this article includes the following figure supplement(s) for figure 2:

**Figure supplement 1.** Pial, mid-gray, and white-matter surface areas around the visual field.

## Asymmetries in cortical surface area by twin status and gender

The strong correlation between behavior and cortical surface area was computed at the group level, given that the two measures come from different subject groups. Nonetheless, there are independent individual differences in both the behavioral (*Himmelberg et al., 2020*) and cortical surface area (*Benson et al., 2018*) measures. Because the cortical surface area measures come from the HCP dataset, which includes a large number of twin pairs, we were able to assess whether individual measures in surface area asymmetries (HVA and VMA) were shared between twin pairs. Indeed, for monozygotic twin pairs, the intraclass correlation (ICC) of ROI asymmetry in terms of the ROI's angular width is ≥0.5 for most ROIs (*Figure 4*). Interestingly, dizygotic twin pairs have correlations close to 0 for HVA but positive correlations similar to those of the monozygotic twins for the VMA. Age- and gender-matched unrelated pairs have correlations for both HVA and VMA that are not statistically distinct from 0 for all ROIs. In comparison, the correlation of mean V1 thickness for the same monozygotic and dizygotic twins is 0.76 and 0.11, respectively, with 68% CIs across 10,000 bootstraps of 0.67–0.82 and −0.07 to 0.29. For the V1 mid-gray surface area these correlations are 0.90 (68% CI: 0.88–0.93) and 0.71 (68% CI: 0.57–0.81), respectively. The number of twins is not large enough to make a precise estimate of a heritability fraction for asymmetry, as the heritability confidence combines the uncertainty in the monozygotic and the dizygotic data. The general pattern, however, whereby there is a large difference in correlation between monozygotic and dizygotic twins for the HVA, but not for the VMA, suggests a larger heritable component for the HVA, and that the HVA and VMA may be controlled by different mechanisms.

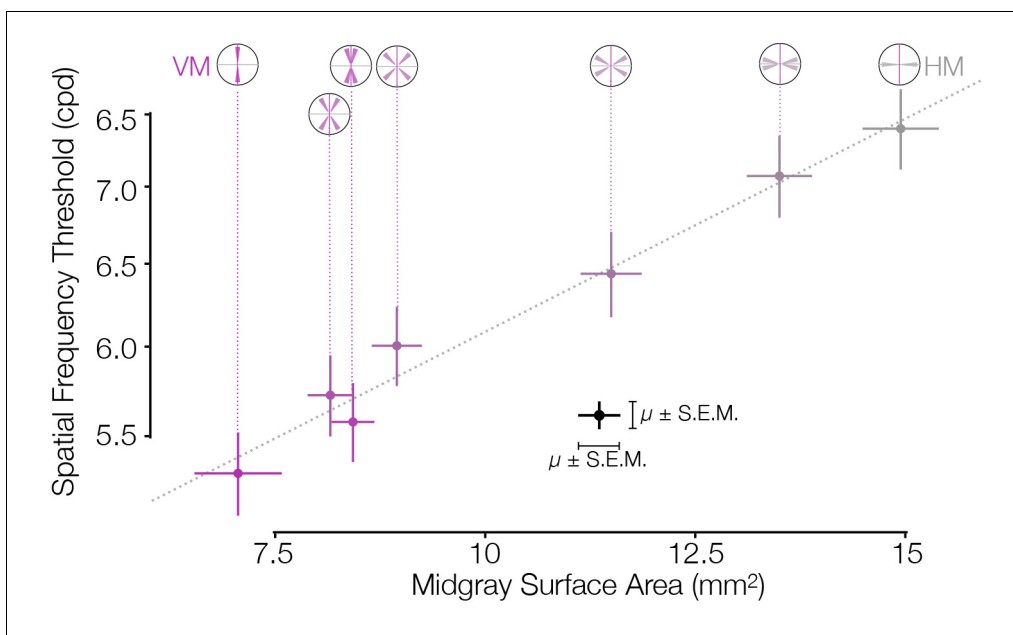

**Figure 3.** Asymmetries in behavior and cortical surface area are correlated in distinct subject groups. V1 mid-gray surface area measured from the HCP dataset (*Benson et al., 2018*) is highly correlated with the spatial frequency thresholds measured by *Barbot et al., 2021* in matched ROIs. Surface areas (x-axis) were computed for ROIs whose visual field positions were limited to 4–5° of eccentricity in order to match the psychophysical experiments. Areas were summed over regions indicated by the visual field insets (and divided by two for all ROIs except the VM and HM, since the latter two ROIs have half the visual field area of the other ROIs). The Pearson correlation is 0.99. HCP, Human Connectome Project; HM, horizontal meridian; ROI, region of interest; VM, vertical meridian.

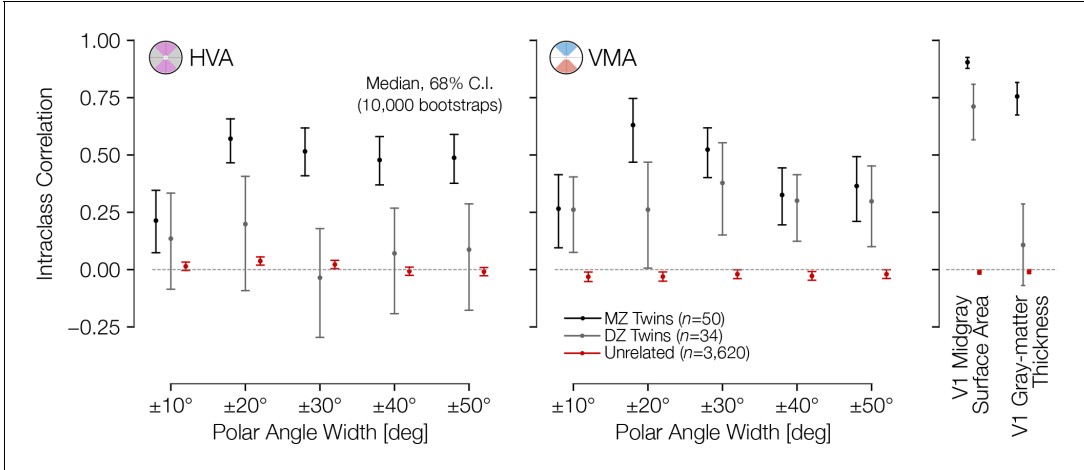

**Figure 4.** Asymmetries in cortical surface area are correlated across twins. Intraclass correlation (ICC) coefficient between twin pairs (monozygotic and dizygotic) and age- and gender-matched unrelated pairs. The correlations were computed for the HVA (left) and VMA (right) for wedge widths, as in *Figure 2B and C*. An unbiased estimate of the ICC was used (see Materials and methods). HVA, horizontal-vertical asymmetry; VMA, vertical meridian asymmetry.

Because cortical surface area differs between males and females, we examined whether the radial asymmetries do as well. We find that the cortical VMA and HVA do not differ substantially by sex or across age-groups (*Appendix 1—figure 1*). This is consistent with psychophysical findings that the HVA and VMA are similar in males and females (*Barbot et al., 2021*; *Purokayastha et al., 2021*). Whereas the average size of V1 is about 10% larger in males, V1 size normalized to the entire cortical surface area is similar for males and females (*Appendix 1—figure 2*).

## Asymmetries in cortical surface area and retinal cell density

Asymmetries around the visual field exist in many parts of the visual system, beginning with the optics (*Jaeken and Artal, 2012*; *Polans et al., 2015*; *Liu and Thibos, 2019*) and the retinal cone density (*Curcio et al., 1990*; *Song et al., 2011*). However, the cortical asymmetries are not simply inherited from the cone photoreceptors. Rather, the asymmetries become larger in measures of RGC density compared to cones, and become substantially larger still in surface area of V1 (*Figure 5*). Below 8° of eccentricity, cone density is greater on the horizontal than vertical meridian, with an HVA of ~20% (*Figure 5A*; *Curcio et al., 1990*). Over the same eccentricity range, the HVA measures up to 40% in midget RGC density and 60–80% in visual cortex. In this eccentricity range, the VMA of the cone density is 0 or slightly negative (*Figure 5B*): the upper visual field (inferior retina) has higher cone density than the lower visual field (superior retina), indicating that cone density cannot be the source of the behavioral VMA (*Kupers et al., 2019*). The VMA becomes positive in midget RGC density (up to ~20%) and still larger in the cortex (~40–100%). Hence, the radial asymmetries are substantially larger in the cortex compared to both the cone photoreceptors and the mRGCs. Note that neither the changes in polar angle asymmetries from cone density to mRGC density nor from mRGC density to cortex are simple scale factors; rather, the differences across stages of processing vary with eccentricity in a manner that suggest new properties at each stage.

This finding that polar angle asymmetries are larger across successive stages of visual processing parallels findings regarding eccentricity: the cone density is greatest in the fovea, falling off with increasing eccentricity (*Curcio et al., 1990*) this fall-off with eccentricity becomes steeper in the midget ganglion cell array (*Curcio and Allen, 1990*; *Dacey, 1993*; *Watson, 2014*), and steeper still in V1 (*Adams and Horton, 2003*). Given that most human visual perception relies on signals passing through V1, the increases in the foveal bias and the radial asymmetries (HVA and VMA) likely have important implications for visual performance. The overall size of V1 and the cortical magnification function (averaged across polar angles) correlate with various measures of perception, including vernier acuity (*Duncan and Boynton, 2003*; *Song et al., 2015*), perceived angular size (*Murray et al.,*

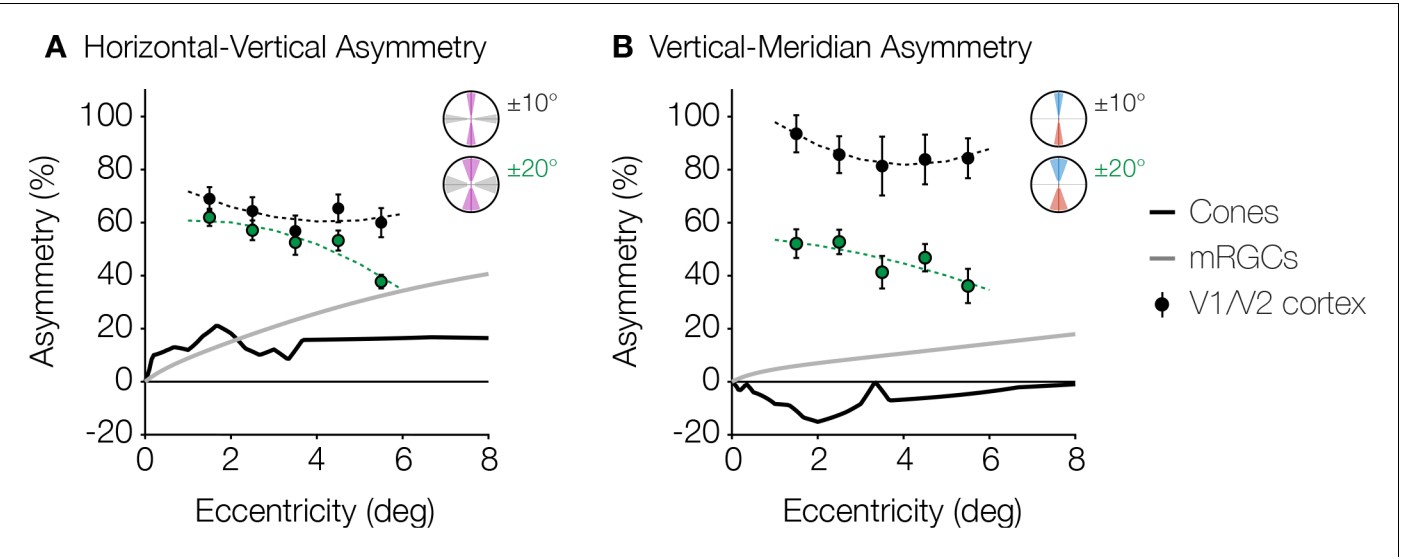

**Figure 5.** Asymmetries from retina and visual cortex. (**A**) HVA and (**B**) VMA for cone density (black line) and mRGC density (gray line) when comparing the cardinal meridians, and V1/V2 cortical surface area within the ±10° ROIs (black markers) and ±20° ROIs (green markers) as a function of eccentricity. Cone density data are in counts/deg² (*Curcio et al., 1990*). Midget RGC densities represent the receptive field counts/deg² using the quantitative model by *Watson, 2014*. V1/V2 cortex data are calculated from the cortical surface area in mm²/deg² within 1° non-overlapping eccentricity bands from 1° to 6° of the ±10° and ±20° ROIs. Markers and error bars represent the median and standard error across bootstraps, respectively. Data are fitted with a second-degree polynomial. HVA, horizontal-vertical asymmetry; RGC, retinal ganglion cell; ROI, region of interest; VMA, vertical meridian asymmetry.

*2006*), and the strength of some illusions (*Schwarzkopf et al., 2011*). The cortical HVA and VMA in individuals may similarly predict radial asymmetries in their visual performance.

## Discussion

Here, we reported cortical magnification at a fine angular resolution around the visual field and in the largest available subject pool, allowing us to measure the organization of cortical magnification with high precision. This precision in turn enabled our comparison between cortical magnification and behavior, between cortical magnification of twins, and between cortical magnification and retinal cell densities. The fine angular resolution is especially important for assessing the VMA. The VMA falls off sharply with distance from the vertical meridian: the size of the VMA within 10° of the vertical meridian is ~80%, whereas within 40° it is only ~10%. Prior studies have also shown radial asymmetries in visual cortex, in fMRI BOLD amplitude (*Liu et al., 2006*; *O'Connell et al., 2016*), in cortical volume (*O'Connell et al., 2016*), in cortical magnification in human (*Silva et al., 2018*) and Macaque monkey (*Van Essen et al., 1984*), in pRF size in human (*Silva et al., 2018*), in spatial frequency preference in human (*Aghajari et al., 2020*), and in diffusion MRI in human (*O'Connell et al., 2016*). For instance, *O'Connell et al., 2016* found that a greater volume of visual cortex is activated by lower than upper hemifield stimuli. *Silva et al., 2018* quantified cortical magnification in visual field quadrants (lower, upper, left, and right) for V1–V3 and reported a greater magnification in the horizontal than the vertical quadrants and in the lower than the upper quadrant. These reports agree broadly with our measurements. However, neither behavior (*Carrasco et al., 2001*; *Cameron et al., 2002*; *Abrams et al., 2012*; *Baldwin et al., 2012*; *Barbot et al., 2021*) nor cortical magnification is uniform within quadrants (*Figure 2*). *Silva et al., 2018* additionally quantified the pRF size and cortical magnification of small polar angle bins, and observed a general trend for pRFs to be smaller and for magnifications to be larger on the horizontal than the vertical. However, they observed that their finely binned cortical magnification data were noisy, especially near the vertical, and it is unclear whether the magnification measurements of their polar angle bins in V1 are statistically distinct from each other (*Appendix 1—figure 3*; see also Figure 4B in *Silva et al., 2018*). Hence assessing the

relation between cortical area and behavior requires measurement at a fine angular resolution and in a large dataset.

In this study, we chose to emphasize cortical surface area as our measure of cortical asymmetry rather than pRF size or BOLD amplitude. Whereas all of these measures reflect important properties of visual system organization, we consider surface area, being closely tied to anatomy, to provide the most direct interpretation and to be most invariant to changes in experimental details. BOLD amplitude is influenced by a variety of non-neural factors, such as vascular density (*Lee et al., 1995*; *Winawer et al., 2010*; *Kay et al., 2019*) and cortical surface orientation relative to the MRI bore (*Gagnon et al., 2015*). PRF size is dependent on both neural receptive field size and the scatter of neural receptive field centers within a voxel (*Amano et al., 2009*). The within-voxel scatter is influenced by cortical magnification, such that regions with lower magnification, like the vertical meridian, would have a greater change in neural RF center across the voxel, and hence potentially larger pRFs even if neural RFs were matched. In contrast, pRF size is unlikely to influence surface area or magnification estimates, since these estimates are computed from pRF centers only. Moreover, pRF size can be influenced by stimulus pattern (*Le et al., 2017*) and task (*Kay et al., 2015*; *Klein et al., 2014*).

Although variation in performance with respect to polar angle is less well studied than variation with respect to eccentricity, the size of the polar angle asymmetries in behavior is substantial. At 6° of eccentricity, the VMA and HVA in contrast sensitivity are >40% (*Figure 1C*, 0° vs. dotted-line; *Figure 1E*, upper vs. lower at 0°). Additionally, the shapes of these performance fields are resilient. For example, both the HVA and VMA are present for stimuli of different orientation, size, luminance, eccentricity, and spatial frequency and under binocular and monocular viewing conditions (*Carrasco et al., 2001*; *Cameron et al., 2002*; *Montaser-Kouhsari and Carrasco, 2009*; *Abrams et al., 2012*; *Baldwin et al., 2012*; *Corbett and Carrasco, 2011*; *Greenwood et al., 2017*; *Himmelberg et al., 2020*; *Barbot et al., 2021*). They also remain constant after both endogenous (voluntary) (*Purokayastha et al., 2021*) and exogenous (involuntary) (*Carrasco et al., 2001*; *Cameron et al., 2002*) covert attention have exerted their effects on perception. That is, attention improves performance in a variety of tasks at all polar angles to a similar degree, for example, orientation discrimination and acuity, but does not diminish the radial asymmetries. Furthermore, the shape of performance fields is maintained in visual short-term memory (VSTM), implying that the quality with which a visual stimulus is perceived and encoded at different locations affects the quality with which it is maintained in VSTM (*Montaser-Kouhsari and Carrasco, 2009*). The results reported here—differential resources devoted to the different meridians in V1—may underlie both the perceptual and VSTM phenomena.

Previous work examining the visual cortex in the context of twins and heritability has found some evidence of genetic control. The variance of resting-state fluctuations within the visual cortex has a strong genetic component (*Yang et al., 2016*), and the surface areas of V1–V3 are more correlated for monozygotic than dizygotic twins (*Alvarez et al., 2021*; *Benson et al., 2021b*). Interestingly, our analysis of the asymmetries of twin pairs in the dataset demonstrated differences between HVA and VMA. In all but the ±10° ROIs, monozygotic twins had significantly higher correlations in HVA than dizygotic twins or unrelated pairs (*Figure 4*). This finding is consistent with a strong genetic effect on HVA, a finding which might reflect a tendency for genetic factors to influence the topography of visual cortex more generally. Surprisingly, however, the twin-pair correlations of the VMA were similar for monozygotic and dizygotic twins. This might be due to measurement limitations (the VMA is estimated from half as much data per subject as the HVA) or it might indicate a separate mechanism from HVA that is more influenced by environmental factors. It is unlikely that measurement noise is the sole explanation because we found a positive correlation for both monozygotic and dizygotic twins for the VMA, whereas noise would tend to reduce correlations.

To conclude, this study reveals a tight link between human perception around the visual field and the anatomical organization of V1: both cortical magnification and visual task performance are greatest on the horizontal meridian, lesser on the lower vertical meridian, and the lowest on the upper vertical meridian. Both the behavioral and cortical HVA/VMA fall off gradually with angular distance from the cardinal meridians, with both performance and magnification reaching equality at the intercardinal angles. Moreover, we show that these cortical asymmetries reflect the large-scale organization of the visual pathways, becoming more stark at multiple successive stages, and that they are highly correlated between monozygotic twin pairs. These findings show a clear relation between

biological structure and behavior and reveal how an increasingly large radial asymmetry across the visual system contributes to visual perception. Comprehensive models of vision must take into account not only performance and anatomical variations across eccentricity, but also variations as a function of polar angle across iso-eccentric locations.

## Materials and methods

### Ethics statement

No human subjects data were collected for this paper. All data used in this paper were obtained from previous publications and publicly available datasets. Primarily, analyses were performed using data from the HCP (*Van Essen et al., 2012*), including data from the HCP that were reanalyzed by subsequent studies (*Benson et al., 2018*; *Benson et al., 2021b*). Additionally, *Figures 1* and *3* include data replotted from previous publications by the authors (*Carrasco et al., 2001*; *Abrams et al., 2012*; *Barbot et al., 2021*), and *Figure 5* includes publicly available data from *Curcio et al., 1990*. In all cases, informed consent was obtained from subjects in the original studies, and all applicable use policies were followed in the use of the data. No personal health information is included in this paper or in the associated dataset or code.

### Code and data availability

All data analyzed in this project was previously available from other sources. The results of all analyses and the source code required to produce them, as well as documentation, are publicly available at an Open Science Framework website (https://osf.io/5gprz/; DOI:10.17605/OSF.IO/5GPRZ) with the exception of analyses that rely on restricted data from the HCP, in which case code that will reproduce the analyses given the restricted data is provided. This paper's Github repository can be found at https://github.com/noahbenson/performance-fields/ (copy archived at swh:1:rev:81daa65b3bc69a085acab9508610f853f6a7dcb0, *Benson, 2021a*).

### Data Sources

Structural preprocessed data were obtained from the HCP Young Adult dataset (https://db.human-connectome.org/) (*Van Essen et al., 2012*). Genotype data for these subjects (*i.e.*, twin pairings) were also obtained from the HCP. Retinotopic maps were obtained from the HCP 7 Tesla Retinotopy Dataset (*Benson et al., 2018*). Manually drawn iso-polar angle and iso-eccentricity contours were obtained from *Benson et al., 2021b*. Behavioral data were obtained from *Abrams et al., 2012* and *Barbot et al., 2021*. Behavioral data were obtained from a separate set of subjects than the structural and retinotopic mapping data because we have no access to the HCP subjects to administer behavioral experiments. The HCP includes some simple acuity measurements for the subjects, but these measures are not positioned around the visual field.

### Cortical calculations

We used the HCP 7 Tesla Retinotopy Dataset (*Benson et al., 2018*) ($n$ = 163 subjects) to characterize the cortical HVA and VMA. We computed the percent asymmetry in cortical surface area for V1 (HVA) or V1/V2 (VMA) using the formula $\Delta(a_1, a_2) = 100 \times (a_1 - a_2)/\text{mean}(a_1, a_2)$, where $a_1$ and $a_2$ are the surface areas of the matched horizontal and vertical ROIs or the matched upper and lower ROIs. To test the angular specificity of these asymmetries, we defined several wedge-shaped ROIs surrounding each cardinal meridian, ranging in width from narrow to wide (±10° to ±50° of polar angle from the meridians). All ROIs were restricted to 1–6° of eccentricity.

ROIs were computed in several steps. Initially, V1 and V2 were divided into six coarse-grain sectors using manually drawn iso-polar angle and iso-eccentricity contours as boundaries (*Benson et al., 2021b*). These sectors were bounded in the tangential direction by either the UVM and the HM or by the HM and the LVM. In the radial direction, they were bounded by either 1–2°, 2–4°, or 4–7° of eccentricity. These 24 sectors (six per visual area per hemisphere) were then resampled in each subject into a smaller set of sectors corresponding to the ROIs mentioned above. Using the notation in which the UVM, the HM, and the LVM have polar angles of 0°, 90°, and 180°, respectively, these resampled sectors were bounded, in the tangential direction, by 0–10°, 10–20°, 20–30°, and so on, up to 170–180° of polar angle and, in the radial direction, by 1–2°, 2–3°, 3–4°, 4–5°, and 5–

6° of eccentricity. These individual fine-grain sectors were then combined into the various ROIs analyzed in this study.

Resampling the coarse-grain sectors published by *Benson et al., 2021b* into fine-grain sectors was performed by resampling each coarse-grain sector independently. To illustrate the procedure, we describe an example coarse-grain sector bounded by 1–2° of eccentricity and by the HM and the UVM. This coarse-grain sector would be divided into 18 fine-grain sectors, each 10° of polar angle wide (but it would not need to be subdivided further in terms of eccentricity). To ensure that the fine-grain sectors are contiguous and to reduce the effects of measurement noise, we drew fine-grain sector boundaries using a measurement of the distance between the two coarse-grain polar angle boundaries (the HM and the UVM in this case). The distance metric used was $r = (d_1 - d_2)/(d_1 + d_2)$, where $d_1$ and $d_2$ are the distances in mm along the cortical surface from each of the respective boundaries (the UVM and HM, respectively, in this case). Note that $r$ is always $-1$ at one of the two boundaries, is one at the other boundary, and varies smoothly between them. Given this distance metric, we then found the values of $r$ at which the fine-grain sector polar angle boundaries occur. To mitigate noise in the polar angle measurements, we employed isotropic interpolation from the scikit-learn toolbox (*Pedregosa, 2011*), which yields a bijective continuous relation between $r$ and polar angle that minimizes the prediction error of the polar angles in terms of $r$. Using this relation we determined the values of $r$ that produced the desired subdivisions of the coarse-grain sector's polar angle values and divided the sector accordingly. Because polar angle and eccentricity are effectively orthogonal in V1 and V2, and because each sector was bounded by iso-polar angle lines on two opposite sides and by iso-eccentricity lines on the other two sides, the same resampling procedure was used for the radial and tangential directions. Example ROIs are shown in *Figure 2A*.

A separate set of wedges and eccentricities was used for the data in *Figure 3* that was matched to the behavioral experiment. These behavior-matched polar angle wedges were 15°-wide each and were limited to 4–5° of eccentricity. Explicitly, these ROIs represent 0±7.5°, 15±7.5°, 30±7.5°, 45±7.5°, and so on, all the way around the visual field. Similarly, a separate set of wedges and eccentricities were used for the data in *Appendix 1—figure 3*; these bins were limited to between 1° and 6° of eccentricity and used polar angle bins matched to *Figure 4* by *Silva et al., 2018*. Explicitly, these ROIs represent 0±5.625°, 11.25±5.625°, 22.5±5.625°, 33.75±5.625°, 45±5.625°, and so on, all the way around the visual field. Note that a weakness of the ROI resampling method is that it does not explicitly handle ipsilateral representations of the visual field on cortex. Such representations, according to a previous analysis of the HCP dataset used here, are common near the LVM and to a lesser extent near the UVM on the V1/V2 border (*Gibaldi et al., 2021*). Our method treats these ipsilateral pRFs as noise around the vertical meridian and lumps them in with the polar angle wedge that is nearest the vertical. When the wedges are small this can result in an overestimation of the vertical meridian surface. Thus, the data point at the LVM should be read and interpreted with caution.

For both the 10°- and the 15°-wide wedges, the surface areas of each ROI (which were calculated from V1 alone) were compared with the areal cortical magnification formula of V1 described by *Horton and Hoyt, 1991*: $m_{HH}(x, y) = (17.3 \text{ mm}/(\rho(x, y) + 0.75°))^2$, where $\rho(x, y)$ is the eccentricity in degrees of the visual field coordinate $(x, y)$. The integral of $m_{HH}(x, y)$ over the visual field representation of an ROI yields the cortical surface area $c_{ROI}$ predicted by Horton and Hoyt for that ROI: $c_{ROI} = \int_{(x,y) \in ROI} m_{HH}(x,y) \, dx \, dy$. Because Horton and Hoyt's formula does not incorporate polar angle, its predictions should fall between the surface areas of the vertical ROIs and the horizontal ROIs. For both wedge widths, this was the case. For the 10°-, 20°-, 30°-, 40°-, and 50°-wide ROIs in *Figure 2B*, Horton and Hoyt predict 2.2, 4.3, 6.5, 8.6, and 10.8 cm$^2$ surface areas, respectively. For the 15°-wide wedges, the predicted surface area for each wedge is 12.8 mm$^2$, which is between the surface areas of the vertical and the horizontal ROIs in *Figure 3*.

ICC values plotted in *Figure 4* were calculated between the 38 monozygotic twins, the 33 dizygotic twins, and the 2858 age- and sex-matched unrelated pairs from the HCP 7 Tesla Retinotopy Dataset (*Benson et al., 2018*) for which manually annotated iso-angle and iso-eccentricity contours were available (*Benson et al., 2021b*). The 2858 unrelated pairs represent the subset of the possible 13,203 unique subject pairs from the 163 retinotopy subjects that are not twins or otherwise related, that are of the same gender, and that are in the same HCP age-group. The monozygotic twins included 15 male twin pairs and 23 female twin pairs while the dizygotic twins included 17 male and 16 female twins. Precise ages are not available for subjects, but all subjects in the HCP Young Adult Connectome were 22–40 years old. Error bars are plotted as the 16th and 84th percentiles across

10,000 bootstraps. The V1 mid-gray surface area and V1 gray-matter thickness correlations in *Figure 4* were calculated using the manually labeled V1 boundaries (limited to 0–7° of eccentricity); the average thickness of this V1 ROI was used. The formula used for the ICC is an unbiased estimate derived from the one-way ANOVA (*McGraw and Wong, 1996*): $\hat{r}_{\mathrm{ICC}} = (\mathrm{MS_r} - \mathrm{MS_w})/(\mathrm{MS_r} + \mathrm{MS_w})$, where $\mathrm{MS_r}$ and $\mathrm{MS_w}$ are the mean-square of the across-pairs factor and the residuals, respectively. For a $2 \times n$ matrix **X** of twin-pair measurements such that $x_{1,j}$ is the measurement of the first twin of twin pair $j$ and $x_{2,j}$ is the measurement of the second twin, then $\mathrm{MS_r}$ and $\mathrm{MS_w}$ are given by *Equations 1 and 2*.

$$\mathrm{MS}_r = \frac{1}{2(n-1)} \sum_{j=1}^{n} (\bar{x}_{.j} - \bar{x}) \tag{1}$$

$$\mathrm{MS}_w = \frac{1}{n} \sum_{i \in \{1,2\}} \sum_{j=1}^{n} (x_{i,j} - \bar{x}_{.j}) \tag{2}$$

$$\text{where } \bar{x}_{.j} = \frac{x_{1,j} + x_{2,j}}{2},$$

$$\text{and } \bar{x} = \frac{1}{2n} \sum_{i \in \{1,2\}} \sum_{j=1}^{n} x_{i,j}$$

## Retinal calculations

The raw data used for *Figure 5* were previously published or from publicly available analysis toolboxes. Cone density data were extracted from post-mortem retinal tissue of eight human retinae published by *Curcio et al., 1990* for 0–8° eccentricities (step size 0.1°) and summarized in the ISET-BIO toolbox (https://github.com/isetbio/isetbio) (*Cottaris et al., 2019*). Midget RGC (mRGC) densities were extracted from the analytic model by *Watson, 2014*. Cortical asymmetries were computed from the HCP 7 Tesla Retinotopy Dataset (*Benson et al., 2018*) as percent difference in V1/V2 cortical surface area in ±10° and ±20° wedges centered on the cardinal meridians for 1° bins between 1° and 6° eccentricity. The ±10° and ±20° wedges were identical to those used in *Figure 2*, and both were plotted due to the possibility of bias introduced by the ipsilateral cortical representation near the vertical meridian discussed in the previous section (see Cortical calculations). Median and standard error of cortical surface area were computed from bootstrapped data across subjects (1000 iterations).

Meridian asymmetries in cone density and mRGC density were defined as percent change in density values at the cardinal meridians (for retina) or V1/V2 cortical surface area within polar angle wedges centered on cardinal meridians as follows:

$$\mathrm{HVA} = 100 \frac{\mathrm{mean}(nasal, temporal) - \mathrm{mean}(superior, inferior)}{\mathrm{mean}(nasal, temporal, superior, inferior)}$$

$$\mathrm{VMA} = 100 \frac{superior - inferior}{\mathrm{mean}(superior, inferior)}$$

## Statistical analysis and measurement reliability

This manuscript does not assess statistical significance. CIs computed for *Figures 1* and *3* are the standard error of the mean. CIs for *Figures 2* and *5* are the standard deviation of the median across bootstraps. CIs for *Figure 4* are the 16th and 84th percentiles across 10,000 bootstraps.

Although the behavioral data reanalyzed here were published previously, we have included an overview of their quantification as well. *Abrams et al., 2012* measured 75%-correct contrast thresholds in a 2-AFC orientation discrimination task. Participants reported the orientation (±15° of vertical) of Gabor targets (6 cycles/deg) presented at different iso-eccentric locations at 6° of eccentricity. Repeated-measures ANOVAs were used to assess changes in contrast sensitivity as a function of visual hemifield (upper vs. lower) and angular distance from the VM (0°, 15°, 30°, 45°, 67.5°, and 90°). Paired *t*-tests were corrected for multiple comparisons using a Bonferroni correction. (*Barbot et al.,*

*2021*) measured orientation discrimination performance as a function of the spatial frequency and angular position of high-contrast grating stimuli presented at 10° of eccentricity. Spatial frequency acuity thresholds were computed from psychometric fits as the stimulus spatial frequency at which performance was 75% correct in the 2-AFC (±45°) orientation discrimination task. Linear mixed-effects models were used to assess the changes in visual acuity between upper and lower hemifields as a function of the angular distance from the VM (0–90°, with 15° steps).

The wedge-shaped ROIs around the visual field were calculated by subdividing visual sectors whose boundaries were hand-drawn on V1 and V2. These sectors were each drawn by four trained raters, and the reliability of these boundaries was assessed extensively in the source publication (*Benson et al., 2021b*). To summarize: these boundaries are incredibly reliable, with differences between raters accounting for <5% of all variance across subjects and raters. The ROIs were additionally informed by the pRF polar angle and eccentricity measurements from the HCP 7 Tesla Retinotopy Dataset (*Benson et al., 2018*). Again, reliability of these measurements was discussed extensively in the original paper and is high across independent fMRI scans.

## Acknowledgements

The authors thank Antonio Fernández, Michael Jigo, and Drs. Marc Himmelberg and Jan Kurzawski for helpful comments. This work was supported by NIH NEI RO1-EY027401 to MC and JW as well as a Data Science Environments project award from the Gordon and Betty Moore Foundation (Award #2013-10-29) and the Alfred P Sloan Foundation (Award #3835) to the University of Washington eScience Institute. Data were provided in part by the Human Connectome Project, WU-Minn Consortium (Principal Investigators: David Van Essen and Kamil Ugurbil; 1U54MH091657) funded by the 16 NIH Institutes and Centers that support the NIH Blueprint for Neuroscience Research, and by the McDonnell Center for Systems Neuroscience at Washington University.

## Additional information

### Competing interests

Marisa Carrasco: Reviewing editor, *eLife*. The other authors declare that no competing interests exist.

### Funding

| Funder | Grant reference number | Author |
| --- | --- | --- |
| National Eye Institute | R01-EY027401 | Marisa Carrasco Jonathan Winawer |
| Gordon and Betty Moore Foundation | #2013-10-29 | Noah C Benson |
| Alfred P. Sloan Foundation | #3835 | Noah C Benson |

The funders had no role in study design, data collection and interpretation, or the decision to submit the work for publication.

### Author contributions

Noah C Benson, Data curation, Software, Formal analysis, Validation, Investigation, Visualization, Methodology, Writing - original draft, Writing - review and editing; Eline R Kupers, Software, Formal analysis, Investigation, Visualization, Writing - review and editing; Antoine Barbot, Formal analysis, Investigation, Writing - review and editing; Marisa Carrasco, Conceptualization, Supervision, Funding acquisition, Writing - original draft, Project administration, Writing - review and editing, Resources; Jonathan Winawer, Conceptualization, Resources, Supervision, Funding acquisition, Writing - original draft, Project administration, Writing - review and editing

## Author ORCIDs

Noah C Benson (iD) https://orcid.org/0000-0002-2365-8265
Eline R Kupers (iD) http://orcid.org/0000-0002-4972-5307
Antoine Barbot (iD) https://orcid.org/0000-0002-3301-4279
Marisa Carrasco (iD) https://orcid.org/0000-0002-1002-9056
Jonathan Winawer (iD) https://orcid.org/0000-0001-7475-5586

## Ethics

Human subjects: No human subjects data were collected for this paper. All data used in this paper were obtained from previous publications and publicly-available datasets in which subjects provided informed consent. Primarily, analyses were performed using data from the HCP (DC Van Essen et al. 2012, Neuroimage 62:2222-2231), including data from the HCP that were reanalyzed by subsequent studies (Benson et al. 2018, J Vis 18:23; Benson et al. 2021, bioRxiv 10.1101/2020.12.30.424856). Additionally, Figures 1 and 3 includes data replotted from previous publications by the authors (Carrasco et al. 2001, Spat Vis 15:61-75; Abrams et al. 2012, Vision Res 52:70-78; Barbot et al. 2021, J Vis 21:2), and Figure 5 includes publicly available data from Curcio et al. (1990, J Comp Neurol 292:497-523). In all cases, informed consent was obtained from subjects in the original studies, and all applicable use policies were followed in the use of the data. No personal health information is included in this paper or in the associated dataset or code.

## Decision letter and Author response

Decision letter https://doi.org/10.7554/eLife.67685.sa1
Author response https://doi.org/10.7554/eLife.67685.sa2

# Additional files

## Supplementary files

• Transparent reporting form

## Data availability

All source code and data have been permanently archived on the Open Science Framework with https://doi.org/10.17605/OSF.IO/5GPRZ.

The following dataset was generated:

| Author(s) | Year | Dataset title | Dataset URL | Database and Identifier |
|---|---|---|---|---|
| Benson NC, Kupers ER, Barbot A, Carrasco M, Winawer J | 2020 | Visual Performance Fields | https://doi.org/10.17605/OSF.IO/5GPRZ | Open Science Framework, 10.17605/OSF.IO/5GPRZ |

The following previously published datasets were used:

| Author(s) | Year | Dataset title | Dataset URL | Database and Identifier |
|---|---|---|---|---|
| Benson NC | 2018 | The Human Connectome Project 7 Tesla Retinotopy Dataset | https://doi.org/10.17605/OSF.IO/BW9EC | Open Science Framework, 10.17605/OSF.IO/BW9EC |
| Van Essen DC | 2012 | The Human Connectome Project Young Adult Dataset, 1200 Subject Release | https://db.humanconnectome.org/ | Amazon S3, s3://hcp-openaccess/HCP_1200 |
| Curcio CA | 1990 | Human photoreceptor topography | https://github.com/isetbio/isetbio | Available as part of the ISETBIO toolbox, github.com/isetbio/isetbio |
| Benson NC | 2021 | Variability of V1, V2, and V3 in a Large Sample of Human Observers | https://doi.org/10.17605/OSF.IO/GQNP8 | Open Science Framework, 10.17605/ |

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

## Appendix 1

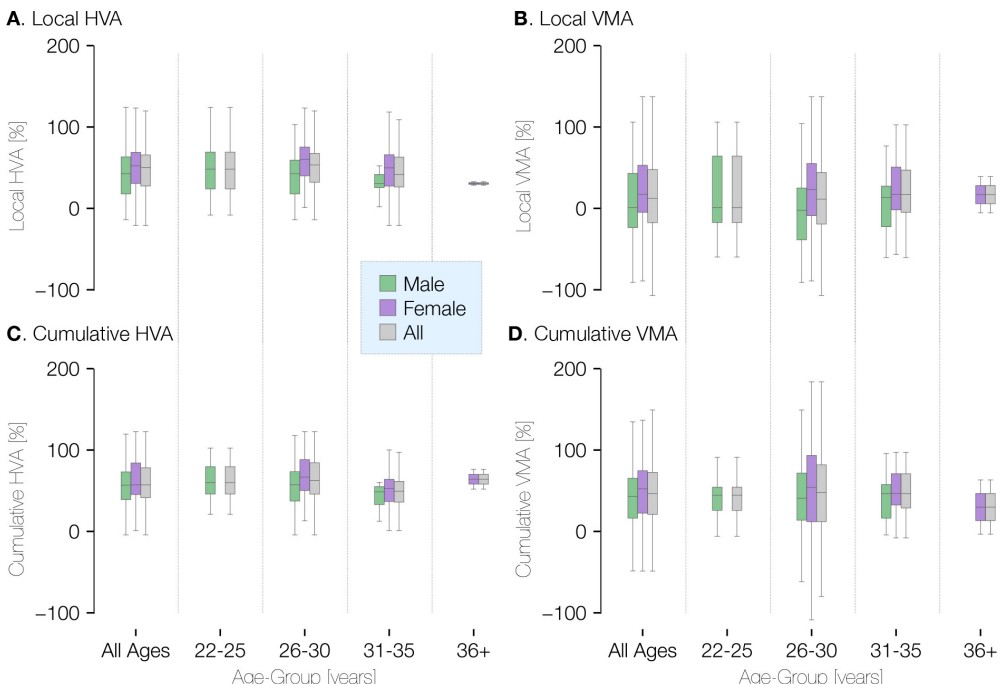

**Appendix 1—figure 1.** Visual field asymmetry in terms of age and gender at ±20˚ of polar angle. Plots of the local (top row) and cumulative (bottom row) asymmetry across subjects in terms of age and gender. The percent horizontal-vertical asymmetry (HVA: horizontal ROI surface area minus vertical ROI surface area, divided by the mean surface area across horizontal and vertical ROIs) is shown in the left column, and the vertical meridian asymmetry (VMA: upper vertical ROI surface area minus lower vertical ROI surface area, divided by the mean surface area across upper and lower ROIs) is shown in the right column. Box-plots show the median (central horizontal line), quartiles (shaded box), and ±95% percentiles (lines).

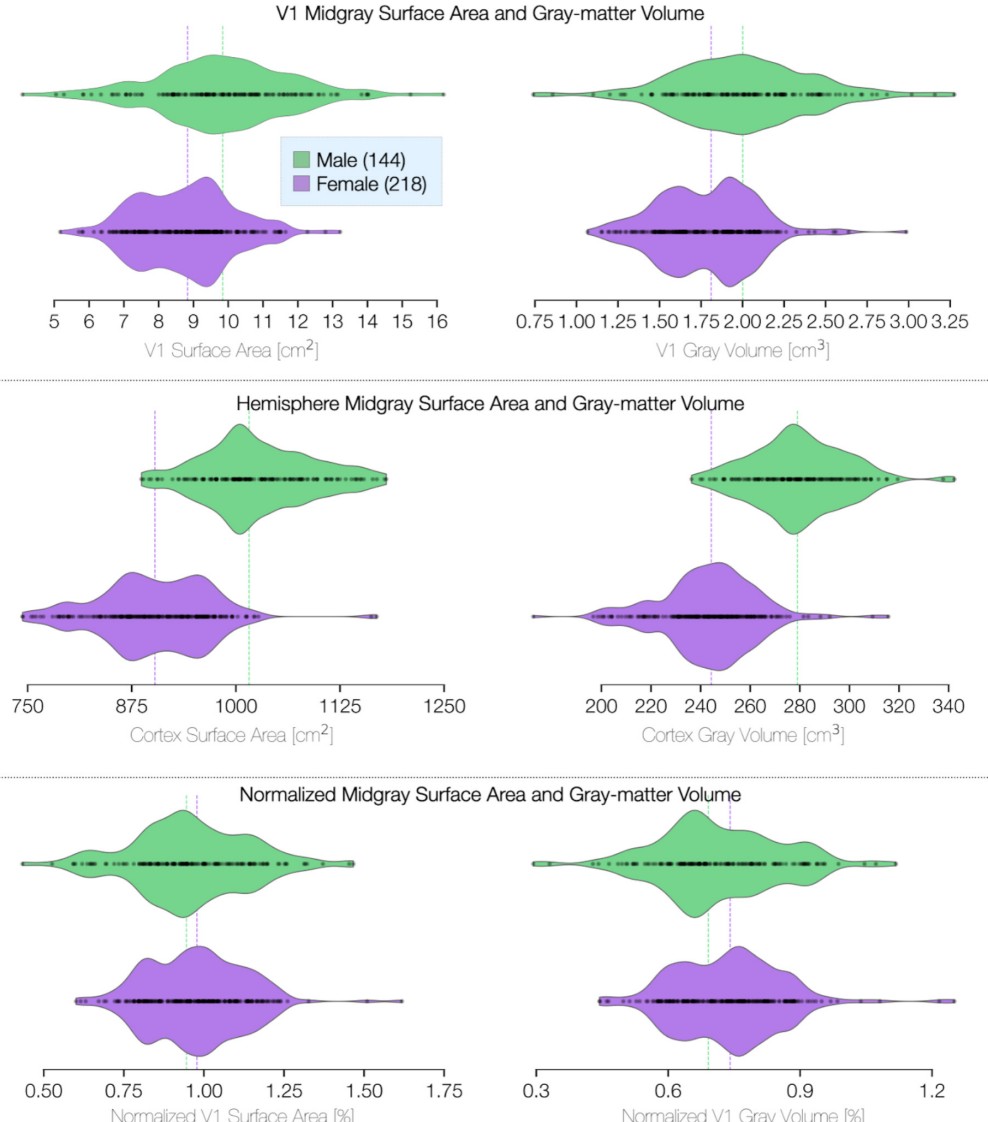

**Appendix 1—figure 2.** V1 size and volume differences in terms of sex. ROI is limited to V1 between 1° and 6° of eccentricity. Violin plots terminate at the exact extrema of the dataset. Surface area was calculated on the mid-gray surface. Volume calculations include the entire gray-matter layer. Whole-hemisphere calculations for surface area and volume exclude the region corresponding to the corpus callosum.

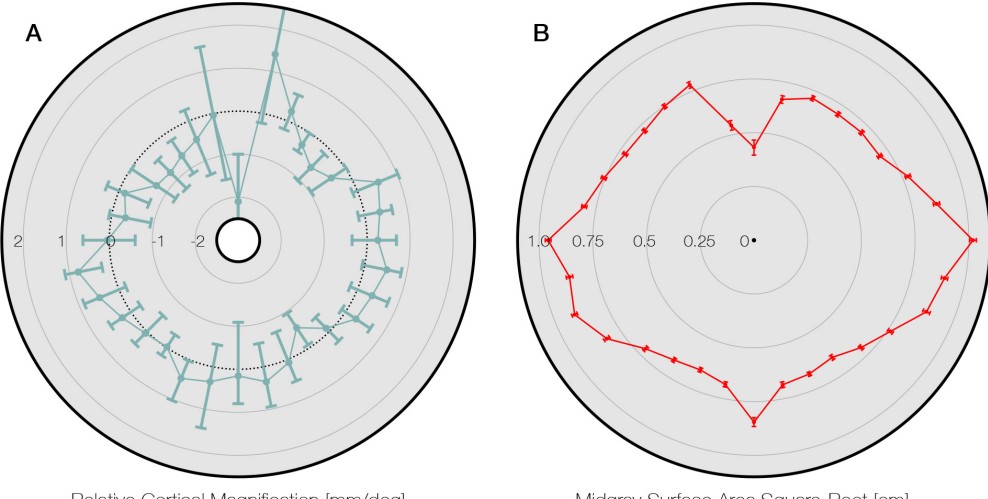

**A** Relative Cortical Magnification [mm/deg]

**B** Midgray Surface Area Square-Root [cm]

**Appendix 1—figure 3.** Comparison to *Silva et al., 2018*. (**A**) Data from *Silva et al., 2018*; *Figure 4*, replotted around the visual field. The gray area of the panel indicates the axis limits of the original plot. Plot points and confidence intervals represent the linear cortical magnification of each polar angle bin after regressing out the effect of eccentricity on cortical magnification. (**B**) The square root of the mid-gray surface area of the 163 subjects analyzed in this paper, plotted for polar angle bins matched to those in panel A. Bins were limited to between 1° and 6° of eccentricity. The square-root of the surface area is used because it should scale linearly with the linear cortical magnification values plotted by *Silva et al., 2018* and replotted in panel A. Notably, the LVM has a substantially higher surface area in panel B than the points around it. This is likely due to the prevalence of ipsilateral pRFs near the LVM (*Gibaldi et al., 2021*). Our method does not explicitly account for ipsilateral representation, and thus parts of cortex with ipsilateral pRFs will tend to be counted as part of the vertical meridian when the polar angle bins are sufficiently small (see also *Methods*). Thus the data point at the LVM should be read and interpreted with caution. pRF, population receptive field. LVM, lower vertical meridian.

