## [Decision Letter]

**Acceptance summary:**

We believe that this manuscript would be of broad interest to readers in the field of human vision and its cortical topography as well as behavioral and genetic links. The investigation of the neurobiological basis of visual task performance asymmetries represents an important contribution to our understanding of how visual system architecture shapes perception. The key claims of the manuscript are well supported by the data, and the approaches used are thoughtful and rigorous.

**Decision letter after peer review:**

Thank you for submitting your article "Cortical Magnification in Human Visual Cortex Parallels Task Performance around the Visual Field" for consideration by *eLife*. Your article has been reviewed by 3 peer reviewers, including Ming Meng as the Reviewing Editor and Reviewer #1, and the evaluation has been overseen Joshua Gold as the Senior Editor. The following individuals involved in review of your submission have agreed to reveal their identity: Geoffrey K. Aguirre (Reviewer #2); Xi-Nian Zuo (Reviewer #3).

Essential revisions:

1. Clarify the relationship with previous findings.

Firstly, the present results demonstrate that 45 deg angles are relatively under-represented on the cortical surface, but without a concomitant decline in perceptual performance (Figure 3). While a prior study demonstrated radial asymmetries in cortical magnification (Silva 2018), that prior study did not have the power and resolution to show the clear reduction in surface area around 45 deg that is shown here. This result is one of the more novel findings of the current study, but is not discussed. It is noted that in the Barbot (2021) paper, acuity was tested with a grating that was oriented at 45 deg. Could this property of the stimulus interact with the radial orientation bias that has been shown in perception and cortical response (e.g., Sasaki 2006).

Secondly, although the results of twins are very impressive and novel from previous findings such as Silva et al., 2018, we suggest that these twins results should be more thoroughly discussed. For example, the correlations in the MZ twins may not be an independent source of information. That is, could a conclusion be drawn from the correlation in the MZ twins that there is a genetic influence specifically for radial asymmetry of the visual cortex? Instead, there may be genetic influences upon the general shape, folding, and functional organization of the cortex as a whole, of which the visual cortex is just one part. It would be informative, for example, if the correlation in MZ twins for visual cortex radial asymmetry is GREATER than the correlation that is observed for any other cortical property (Chen 2013). While we understand that perceptual data are not available for the twin cohort, it may be helpful to describe any relevant prior work on perceptual variation in twin populations (if such measurements exist).

Thirdly, the Silva 2018 paper included a more "fine scale" analysis of cortical magnification as a function of polar angle (Figure 4B). The error bars in this prior report are an order of magnitude larger than in the current measurements, but it would be helpful to have an evaluation of the degree to which the current measures agree with this prior work.

Fourthly, These is an interesting way of discussing the present findings of the human vision system by looking them at the level of the global brain system (e.g., connectomics), for example, how these vision-related heritable features are related to or implicated for their connectome-level findings (Yang et al., 2016, https://pubmed.ncbi.nlm.nih.gov/26891986)?

Lastly but not least, Credit HCP data use: Please visit https://www.humanconnectome.org/study/hcp-young-adult/document/hcp-citations

References:

Sasaki, Yuka, et al. "The radial bias: a different slant on visual orientation sensitivity in human and nonhuman primates." Neuron 51.5 (2006): 661-670.

Chen, Chi-Hua, et al. "Genetic topography of brain morphology." Proceedings of the National Academy of Sciences 110.42 (2013): 17089-17094.

Silva, Maria Fatima, et al. "Radial asymmetries in population receptive field size and cortical magnification factor in early visual cortex." NeuroImage 167 (2018): 41-52.

Yang Z, Zuo XN, McMahon KL, Craddock RC, Kelly C, de Zubicaray GI, Hickie I, Bandettini PA, Castellanos FX, Milham MP, Wright MJ. Genetic and Environmental Contributions to Functional Connectivity Architecture of the Human Brain. Cereb Cortex. 2016 May;26(5):2341-2352.

2. There is the obvious confound that the horizontal meridian is represented in the depth of a sulcus, while the vertical meridian is represented close to the gyral crowns. It is recommended to add some consideration in the methods or discussion of why cortical folding can't account for the current results.

3. The statistical model on repeated measurements: in the present work, there are lots of repeated measurements recorded (e.g., Figure 1, across angular distance and meridian). It is a need of clear and comprehensive description on the statistical methods to be reported in the method part.

4. Measurement reliability: this is a fundamental concept of individual differences, which the present work is based on to assess the link between brain, behavior and genetics. The reliability levels of these measurements should be reported due to the importance of understanding the correlational outcomes. For example, In Figure 3, a surprisingly high correlation was reported (r = 0.96). How we interpret this correlation in terms of the psychometric theory of individual differences. Again, how this correlation was derived from such a setting on the repeated measurements.

Some materials for measurement reliability are listed below:

1. https://pubmed.ncbi.nlm.nih.gov/31253883/

2. https://pubmed.ncbi.nlm.nih.gov/33439234/

3. https://pubmed.ncbi.nlm.nih.gov/33685310/

4. https://pubmed.ncbi.nlm.nih.gov/33685291/

5. The cortical surface representation is described as an "amplification" of asymmetries that are present in the retina. Looking at Figure 5, however, it doesn't appear that a multiplicative scaling of the cone or midget RF functions would fit the cortical data. The cortical asymmetries are certainly larger, but they are of a different form with eccentricity. This might be worth acknowledging, and perhaps considering that perceptual measures as a function of eccentricity and polar angle could deepen the correspondence with the cortical data.

Also, as the authors noted too, "behavioral pattern may vary with task". It would be helpful in general if the relationship between the present cortical magnification finding and behavioral results could be discussed with further details.

6. ICC: should be non-negative. In Figure 4, the negative ICCs appeared for DZ twins for some polar angle widths. Please clarify the reason.

*Reviewer #1 (Recommendations for the authors):*

1. For the results of MZ, DZ twins analysis, I would suggest to provide more information about the included twin participants, such as the numbers of twins, their age ranges, etc.

2. Figure symbols can perhaps be made larger, and avoid using colours that are a little too light.

*Reviewer #2 (Recommendations for the authors):*

1. The introduction contains the statement "This eccentricity effect is mediated by a far greater density of photoreceptors and retinal ganglion cells (RGCs) for the fovea than periphery". This should be edited to indicate the CONE photoreceptors.

2. In Figure 2, I understand the motivation for offsetting points on the x-axis to allow them to be more visible, but I found myself wanting to compare slopes, and points of cross-over of the fit lines, and this was frustrated by the x-axis shift.

---

## [Author Response]

Essential revisions:1. Clarify the relationship with previous findings.Firstly, the present results demonstrate that 45 deg angles are relatively under-represented on the cortical surface, but without a concomitant decline in perceptual performance (Figure 3). While a prior study demonstrated radial asymmetries in cortical magnification (Silva 2018), that prior study did not have the power and resolution to show the clear reduction in surface area around 45 deg that is shown here. This result is one of the more novel findings of the current study, but is not discussed. It is noted that in the Barbot (2021) paper, acuity was tested with a grating that was oriented at 45 deg. Could this property of the stimulus interact with the radial orientation bias that has been shown in perception and cortical response (e.g., Sasaki 2006).

The cortical surface areas corresponding to the 45° angles reported in the original submission were indeed puzzling and suggestive of some bias in the data or method. We have performed a substantial re-analysis of the results using a now-published dataset of manually-drawn V1, V2, and V3 boundaries (Benson et al., 2021; DOI: 10.1101/2020.12.30.424856); our method section was also substantially updated to accommodate this dataset and some new calculations. Specifically, there are two main changes in the method. First, we now use hand-drawn boundaries of the lower, upper, and horizontal meridians, and of several iso-eccentricity contours, to identify sectors of the V1 map, rather relying on the boundaries found by an automated atlas fit. Second, we now employ a new and more robust method to carve up these sectors into fine-grained regions. The manually labeled boundaries have high inter-rater reliability and their inclusion eliminates any biases that could have derived from the automated boundary-finding method we had previously employed. This re-analysis leaves our previous findings intact and largely unchanged, and it eliminates the apparent mystery of the surface area of the 45° angle, which is no longer under-represented on cortex relative to behavior (Figure 3).

Regarding the Barbot et al. study, the 45 deg orientation was chosen so that it would not contaminate the psychophysical measures at the vertical and the horizontal meridians, as discriminability would be better along the vertical meridian for orientations off vertical and along the horizontal meridian for orientations off horizontal; it is possible that the performance at the intercardinal locations is better than if they had used 0° or 90° orientation.

Secondly, although the results of twins are very impressive and novel from previous findings such as Silva et al., 2018, we suggest that these twins results should be more thoroughly discussed. For example, the correlations in the MZ twins may not be an independent source of information. That is, could a conclusion be drawn from the correlation in the MZ twins that there is a genetic influence specifically for radial asymmetry of the visual cortex? Instead, there may be genetic influences upon the general shape, folding, and functional organization of the cortex as a whole, of which the visual cortex is just one part. It would be informative, for example, if the correlation in MZ twins for visual cortex radial asymmetry is GREATER than the correlation that is observed for any other cortical property (Chen 2013). While we understand that perceptual data are not available for the twin cohort, it may be helpful to describe any relevant prior work on perceptual variation in twin populations (if such measurements exist).

Our intention in analyzing the asymmetry correlations between twins was not to suggest that there is a genetic influence *limited only* to asymmetry, and we have reworded the discussion to further clarify this point. For context, we have added reports of correlations between other cortical properties [Figure 4].

Thirdly, the Silva 2018 paper included a more "fine scale" analysis of cortical magnification as a function of polar angle (Figure 4B). The error bars in this prior report are an order of magnitude larger than in the current measurements, but it would be helpful to have an evaluation of the degree to which the current measures agree with this prior work.

We have expanded our discussion of this paper in the text and have included a supplemental comparison of their data (as digitized from their Figure 4) with our data (Supplementary file 3, P34).

Fourthly, These is an interesting way of discussing the present findings of the human vision system by looking them at the level of the global brain system (e.g., connectomics), for example, how these vision-related heritable features are related to or implicated for their connectome-level findings (Yang et al., 2016, https://pubmed.ncbi.nlm.nih.gov/26891986)?

We have expanded the Discussion and have included text regarding the previous findings of connectome-level heritability in the visual cortex.

Lastly but not least, Credit HCP data use: Please visit https://www.humanconnectome.org/study/hcp-young-adult/document/hcp-citations

We thank the reviewers for catching this oversight and have included the relevant text in the Acknowledgements.

2. There is the obvious confound that the horizontal meridian is represented in the depth of a sulcus, while the vertical meridian is represented close to the gyral crowns. It is recommended to add some consideration in the methods or discussion of why cortical folding can't account for the current results.

In the original submission we reported only the surface areas of the midgray surface (i.e., the halfway point between pial and white surfaces) as a way to minimize bias of the cortical curvature that might arise on the pial surface (where the gyral crown is expected to have a larger surface area than the sulcal valley) or the white surface (where the opposite is expected). We have now included Figure 2S1 as a supplement to Figure 2 a re-analysis of the data using both the white and pial surface areas as well as the midgray surface area. Whereas surface areas and their ratios vary numerically depending on which surface is used for analysis, the main trends hold for all 3 analyses (white, midgray, pial): there is more surface area for the horizontal than vertical and for the lower vertical than upper vertical. Unsurprisingly, this re-analysis substantially affects the HVA, which depends on the gyral and sulcal surface areas, but only slightly the VMA, which depends only on gyral surface areas. We have also added text in the Results to address this topic.

3. The statistical model on repeated measurements: in the present work, there are lots of repeated measurements recorded (e.g., Figure 1, across angular distance and meridian). It is a need of clear and comprehensive description on the statistical methods to be reported in the method part.

The data referenced in Figure 1, and, in fact, all psychophysical data we analyzed, are from previous publications in which these details were reported, including analysis using linear mixed models. We have now duplicated the relevant details from these publications in the Methods along with relevant reports of the inter-rater reliability of the V1-V2 boundaries on which the surface area calculations were based. This subsection of the Methods is now titled Statistical Analysis and Measurement Reliability [P21–22].

4. Measurement reliability: this is a fundamental concept of individual differences, which the present work is based on to assess the link between brain, behavior and genetics. The reliability levels of these measurements should be reported due to the importance of understanding the correlational outcomes. For example, In Figure 3, a surprisingly high correlation was reported (r = 0.96). How we interpret this correlation in terms of the psychometric theory of individual differences. Again, how this correlation was derived from such a setting on the repeated measurements.Some materials for measurement reliability are listed below:1. https://pubmed.ncbi.nlm.nih.gov/31253883/2. https://pubmed.ncbi.nlm.nih.gov/33439234/3. https://pubmed.ncbi.nlm.nih.gov/33685310/4. https://pubmed.ncbi.nlm.nih.gov/33685291/

We have added a section on Statistical Analysis and Measurement Reliability in the Methods section to address the topic of reliability [P21–22]. Additionally, we note that the correlation from Figure 3 is a correlation of mean values across subjects using *different* subject groups for the *x* and *y* axes and thus should not be interpreted as a finding about individual differences. We have clarified this fact in the text.

5. The cortical surface representation is described as an "amplification" of asymmetries that are present in the retina. Looking at Figure 5, however, it doesn't appear that a multiplicative scaling of the cone or midget RF functions would fit the cortical data. The cortical asymmetries are certainly larger, but they are of a different form with eccentricity. This might be worth acknowledging, and perhaps considering that perceptual measures as a function of eccentricity and polar angle could deepen the correspondence with the cortical data.

In this instance we used the word “amplification” loosely to mean that the asymmetries in cortex were consistently higher than the asymmetries in the retina, not in the mathematical sense of a multiplicative scale factor. We have now clarified this in the text, and we have expanded the discussion of this point.

Also, as the authors noted too, "behavioral pattern may vary with task". It would be helpful in general if the relationship between the present cortical magnification finding and behavioral results could be discussed with further details.

We have elaborated on this point in the Results section.

6. ICC: should be non-negative. In Figure 4, the negative ICCs appeared for DZ twins for some polar angle widths. Please clarify the reason.

We have clarified our use of an unbiased estimate of the ICC in the Methods and have provided the formulae for our calculations [Equations 1–2; P20].

Reviewer #1 (Recommendations for the authors):1. For the results of MZ, DZ twins analysis, I would suggest to provide more information about the included twin participants, such as the numbers of twins, their age ranges, etc.

We have expanded our discussion of the twin results substantially and have

provided relevant demographic information.

2. Figure symbols can perhaps be made larger, and avoid using colours that are a little too light.

For Figure 4, we have now increased the marker size. For Figure 2, we plot error bars without any markers. The reason is that the error bars are small, and symbols would obscure the error bars. Nonetheless, because the data points are connected by lines, the values are easy to see and the different series are distinct from one another. We have spent substantial effort configuring the color palette and layout of the figures and have received frequent positive feedback about their appearance. All figures are friendly to readers with common forms of color-blindness, and all data are plotted in vector-formats in order to enable readers to resize the figures freely. Accordingly, we are disinclined to fiddle with their appearance without a specific problem of clarity to address.

However, we have published a Jupyter notebook, included in this project’s original GitHub and OSF repositories, which contains all code for generating the figures as well as documentation and additional details. In particular, the first code cell in this notebook contains a configuration dictionary for the plot colors used in the figures. We invite readers with different aesthetic preferences to edit this dictionary then re-evaluate the relevant sections of the notebook in order to obtain plots with whatever styling they prefer.

Reviewer #2 (Recommendations for the authors):1. The introduction contains the statement "This eccentricity effect is mediated by a far greater density of photoreceptors and retinal ganglion cells (RGCs) for the fovea than periphery". This should be edited to indicate the CONE photoreceptors.

We have fixed this throughout the manuscript.

2. In Figure 2, I understand the motivation for offsetting points on the x-axis to allow them to be more visible, but I found myself wanting to compare slopes, and points of cross-over of the fit lines, and this was frustrated by the x-axis shift.

We have fixed this (Figure 2).